# Extraction and Characterization of Pepsin- and Acid-Soluble Collagen from the Swim Bladders of *Megalonibea fusca*

**DOI:** 10.3390/md21030159

**Published:** 2023-02-27

**Authors:** Chou Mo, Qiaoli Wang, Guangfeng Li, Wanwen Dong, Feng Liang, Chaoxi Wu, Zhiping Wang, Yifei Wang

**Affiliations:** 1Guangdong Provincial Key Laboratory of Advanced Drug Delivery, Guangdong Provincial Engineering Center of Topical Precise Drug Delivery System, Guangdong Pharmaceutical University, Guangzhou 510006, China; 2Department of Cell Biology, College of Life Science and Technology, Jinan University, Guangzhou 510632, China; 3Guangzhou Jinan Biomedical Research and Development Center Co., Ltd., Guangzhou 510000, China; 4Key Laboratory of Innovative Technology Research on Natural Products and Cosmetics Raw Materials, Guangzhou 510632, China; 5Guangdong Provincial Biotechnology Drug & Engineering Technology Research Center, Guangzhou 510632, China

**Keywords:** *Megalonibea fusca* swim bladders, collagen, characterization, structure

## Abstract

There is a growing demand for the identification of alternative sources of collagen not derived from land-dwelling animals. The present study explored the use of pepsin- and acid-based extraction protocols to isolate collagen from the swim bladders of *Megalonibea fusca*. After extraction, these acid-soluble collagen (ASC) and pepsin-soluble collagen (PSC) samples respectively were subjected to spectral analyses and sodium dodecyl sulphate-polyacrylamide gel electrophoresis (SDS-PAGE) characterization, revealing both to be comprised of type I collagen with a triple-helical structure. The imino acid content of these ASC and PSC samples was 195 and 199 residues per 1000 residues, respectively. Scanning electron microscopy demonstrated that samples of freeze-dried collagen exhibited a compact lamellar structure, while transmission electron microscopy and atomic force microscopy confirmed the ability of these collagens to undergo self-assembly into fibers. ASC samples exhibited a larger fiber diameter than the PSC samples. The solubility of both ASC and PSC was highest under acidic pH conditions. Neither ASC nor PSC caused any cytotoxicity when tested in vitro, which met one of the requirements for the biological evaluation of medical devices. Thus, collagen isolated from the swim bladders of *Megalonibea fusca* holds great promise as a potential alternative to mammalian collagen.

## 1. Introduction

Collagen is the most abundant protein in animal tissues, serving as an essential structural component of cartilage, tendons, skin, bone, and other connective tissues and comprising roughly 30% of total animal protein. As the primary extracellular matrix (ECM) protein in humans, collagen accounts for ~33% of total protein and ~75% of the dry weight of the skin [1,2]. Structurally, collagen is composed of three left-handed α chains in a right-handed triple-helical conformation, containing repeating proline-rich Gly-X-Y sequences in which X and Y are most commonly proline and hydroxyproline [3].

Collagen and the hydrolysates derived therefrom exhibit a range of desirable properties, including excellent biocompatibility, limited immunogenicity, and ease of degradation, that have driven their widespread application in the medical, cosmetics, and food industries among others [4,5,6]. Commercially, collagen and its derivatives are mainly isolated by the processing of byproducts collected from cattle, pigs, or other land-dwelling mammals [7]. Concerns about the safety of these collagen sources have been raised, however, owing to outbreaks of zoonotic diseases, including mad cow disease and foot-and-mouth disease [8]. Certain religions or cultural practices can also restrict the use of some of these mammalian collagen products [9,10].Accordingly, there has been growing interest in recent years in leveraging marine organisms as an alternative source of collagen suitable for broad commercial use [11].

*Megalonibea fusca* is a member of the Sciaenidea family that was identified and designated as a new species by the ichthyologist Professor Zhu Yuanding and his team, and it is found in the Taiwan Strait, the East China Sea, and southern portions of the Yellow Sea [12,13]. Magalonibea fusca are now commercially farmed on an industrial scale in many areas in China, with annual outputs in the tens of thousands of tons [14]. Each adult Megalonibea weighs 30–50 kg, on average, with a maximum size as high as 80–100 kg [12]. However, most research related to *Megalonibea fusca* published to date has largely focused on artificial breeding techniques, with other topics remaining largely unaddressed. The swim bladders of these fish contain high levels of collagen, suggesting that they may be an ideal raw material source for collagen extraction. While researchers have previously reported on the extraction of collagen from the swim bladders of other large fish, no such reports have yet been published pertaining to the processing of swim bladders from *Megalonibea fusca*. The present study was thus developed with goal of isolating acid-soluble collagen (ASC) and pepsin-soluble collagen (PSC) samples from these *Megalonibea fusca* swim bladders, as these techniques can avoid compromising the structural integrity of the isolated collagen, as frequently occurs in the context of hot water- or alkali-based extraction techniques. Following their isolation, these collagen samples were further characterized to evaluate their potential as viable marine collagen sources and alternatives to mammalian collagen.

## 2. Results and Discussion

### 2.1. Yields

Relative to dry swim bladder weight, the respective yields of ASC and PSC were 33.38% and 84.79%, suggesting that pepsin was able to markedly enhance extraction efficiency, in line with prior reports [15,16,17]. This is at least partially attributable to the covalent telopeptide crosslinking of collagen molecules, as pepsin can cleave telopeptide residues to enhance overall collagen solubility [18]. Collagen yields relative to dry weight for a variety of previously reported fish species are compiled in Table 1 [15,19,20,21,22,23]. These results suggest species-specific differences in collagen extraction efficiency, with the high yields in the present study indicating that *Megalonibea fusca* swim bladders represent an ideal source of collagen.

### 2.2. SDS-PAGE Separation Results

Next, electrophoretic SDS-PAGE separation patterns were assessed for the prepared protein extracts and for bovine type I collagen, which served as a reference material (Figure 1). This analysis revealed that the extracted collagen in both the ASC and PSC fractions was type I collagen, emphasizing the feasibility of utilizing swim bladders as a robust source of highly pure collagen. The α_1_- and α_2_ chains were present in both the ASC and PSC samples at a ~2:1 ratio, and their dimeric (β-chain) and trimeric (γ-chain) forms were also evident. The molecular weight values for the bands in the PSC samples (α_1_-chain, 134 kDa; α_2_-chain, 120 kDa) were slightly below those in the ASC samples (α_1_-chain, 137 kDa; α_2_-chain, 124 kDa), consistent with pepsin-mediated telopeptide region removal [24].

### 2.3. Amino Acid Analysis

ASC and PSC samples exhibited similar amino acid content, as shown in residues per 1000 total residues in Table 2. Gly was the most abundant amino acid, comprising 35.1% and 34.8% of ASC and PSC samples, respectively, followed by Ala, Pro, and Hyp. This is consistent with the characteristic repeated (Gly-X-Y)_n_ structure of collagen, which is integral to its superhelical stability [25,26]. The imino acid (Pro and Hyp) content in ASC and PSC samples was 195 and 199 residues per 1000 total residues, respectively, with this content having an impact on collagen thermal stability and being impacted by the temperatures at which different species of fish live [27]. Lower Tyr, His, Ile, and Met levels in these samples were consistent with prior reports of type I collagen amino acid content in other fish species [28,29].

### 2.4. UV Absorption Spectrum Analyses

As they contain high levels of aromatic amino acids, including phenylalanine, tyrosine, and tryptophan, proteins generally exhibit a maximum absorbance peak at 280 nm [30]. However, the peak absorbance for collagen samples is closer to 230–240 nm, owing to the low levels of aromatic amino acids in these repetitive proteins [19]. UV spectra for prepared ASC, PSC, and the reference bovine type I collagen samples are shown in Figure 2. All three spectra exhibited a similar absorption peak at approximately 235 nm, with only limited absorption at 280 nm, consistent with the low levels of aromatic amino acids expected in these samples [31].

### 2.5. Fourier Transform Infrared (FTIR) Spectroscopy Analysis

The FTIR spectra for the ASC and PSC samples were largely identical, with characteristic amide A, B, I, II, and III absorption peaks consistent with similar secondary structural characteristics, indicating that pepsin treatment did not interfere with the triple-helical structure of these collagen molecules [32,33]. Absorption peaks at ~3294 cm^−1^ and ~3286 cm^−1^ correspond to the respective N-H stretching vibration absorption peaks for ASC and PSC amide A, confirming the presence of hydrogen bonds [34,35]. Amide B peaks corresponding to asymmetrical CH_2_ stretching were evident at 3076 cm^-1^ and 3072 cm^−1^ [16]. The amide I, II, and III bands have been reported to be associated with the triple-helical structure of collagen fibrils. The amide I peaks of ASC and PSC resulting from the C=O stretching vibrations were evident at 1637 cm^−1^ and 1630 cm^−1^, respectively [36], while respective amide II bands corresponding to the C-N stretching vibrations and the N-H bending vibrations were evident at 1544cm^−1^ and 1537 cm^−1^. Amide III bands were also evident at 1237 cm^−1^ and 1235 cm^−1^, corresponding to in-phase N-H bending vibrations and C-N stretching vibrations [37]. An absorption ratio between the amide III and 1450 cm^−1^ peaks was close to 1, consistent with the triple-helical structure remaining intact [18,38]. The observed spectra were analyzed with the infrared spectroscopic protein Secondary Structure analysis software (Peakfit, version 4.12, SPSS Inc., Chicago, IL, USA), with relative levels of triple-helical collagen being assessed based on the deconvolution of the amide I band (Figure 3 and Figure 4) [39]. This approach revealed that PSC samples exhibited a slightly higher triple-helical proportion (the 1631 cm^−1^/1658 cm^−1^ band ratio) relative to ASC, potentially owing to terminal peptide cleavage during pepsin treatment (Table 3).

### 2.6. Circular Dichroism (CD) Spectrum Analyses

Circular dichroism in generally used for collagens as an indication of intact tertiary structure [40]. Collagen, with a triple-helical structure, generally exhibited a positive CD peak at 221–222 nm and a negative peak at 196–198 nm, with a crossover point at ~213 nm [17,23]. Both ASC and PSC exhibited weak positive peaks at 221 nm, with respective negative peaks at 197 and 196 nm, and respective crossover points of 214 nm and 215 nm. These findings were consistent with the expected results for natural collagen samples in a triple-helical conformation (Figure 5).

### 2.7. X-ray Diffraction (XRD)

XRD analyses are routinely used to assess collagen fiber localization and distribution within mineralized fish tissues [40,41]. The fiber diffraction patterns of freeze-dried ASC and PSC are presented in Figure 6, with the distance between collagen chains corresponding to the first peak (7.33° for ASC and 7.49° for PSC, corresponding to respective computed spacing d values of 12.06 Å and 11.80 Å) [42].The second broad peak (22.56° for ASC and 21.84° for PSC, with corresponding spacing d values of 3.94 Å and 1 Å) for each sample was the result of diffuse scattering, owing to the complex structural characteristics of these collagen samples [43].These observed peaks are consistent with the expected results for collagen samples with a triple-helical structure.

### 2.8. Microstructural Analysis

Next, SEM was used to evaluate the microstructural characteristics of swim bladder (Figure 7A), ASC (Figure 7B), and PSC samples (Figure 7C). While swim bladder tissues exhibited compact aligned fibrous structures, lyophilized ASC and PSC samples exhibited uniform porous structures that were largely similar to one another. Consistent with prior reports, these collagen samples exhibited some degree of surface wrinkling that may be the result of drying-related dehydration [44]. The fibrous and flaky membrane structure of collagen has been shown to be advantageous for cell growth, wound healing, and the formation of new tissue [15].

TEM and AFM analyses further revealed that these ASC and PSC samples were able to self-assemble into elongated collagen fibers (Figure 8 and Figure 9). The D-periodicity of collagen fibrils influences the biological and mechanical properties of collagen matrices [45]. While the ASC samples exhibited a wider average diameter relative to the PSC samples (30 vs. 25 nm), both exhibited similar mean D-periodicity values (69 and 66 nm, respectively). This suggests that these different extraction techniques had a substantial impact on collagen fibril diameter, but only a limited impact on fibril D-periodicity.

### 2.9. Thermal Analysis

The pepsin-mediated removal of terminal peptide regions generally renders PSC samples more sensitive to thermal denaturation than ASC samples, as evidenced by lower T_max_ values [46]. The same was observed in this study, with PSC exhibiting a lower T_max_ value than the ASC samples (Figure 10). The T_max_ peaks of ASC and PSC were observed at 66 °C and 65 °C, suggesting that pepsin digestion had a minor impact on the triple-helical structure of these collagen samples.

### 2.10. Zeta Potential

Zeta potential is an important parameter when analyzing colloid dispersion stability [15,47]. A protein exhibits a net charge of zero at its isoelectric point (pI), defined by the corresponding pH value at which zero net charge is evident [48]. Collagen stability will be impacted by its net surface charge, so zeta potential values for ASC and PSC samples were next analyzed across a range of pH values (2–8). Both ASC and PSC were positively charged in the pH 2–6.5/6.6 range, with respective pI values of 6.5 and 6.6 (Figure 11). This is consistent with previously reported marine collagen pI values ranging from pH levels of 4.71–7.26 [40].

### 2.11. Analyses of Collagen Sample Solubility

ASC and PSC sample solubility was next assessed at a range of pH levels (2–10) and NaCl concentrations (0–6%), as shown in Figure 12. These collagen samples exhibited greater solubility under acidic conditions, such that solubility levels fell as the pH increased within the tested range. This may be attributable to hydrophobic interactions among collagen molecules, with the protein net charge reaching zero only at particular pH values, specifically at the pI [49]. Slight increases in collagen solubility were observed at more basic pH levels, potentially owing to charge repulsion-mediated improvements in protein solubility [50].

When NaCl concentrations were below 2% and 3%, no changes in ASC or PSC solubility, respectively, were observed. Above these concentrations, however, ASC and PSC solubility declined rapidly, consistent with a "salting-out" effect resulting from protein precipitation due to enhanced hydrophobic interactions, interchain aggregation, and ionic salt competition for water [51,52].

### 2.12. Cytotoxicity Analyses

In a CCK-8 assay, neither ASC nor PSC treatment caused any apparent cytotoxicity when used to treat BALB/C-3T3 fibroblasts (Figure 13A) or HaCaT cells (Figure 13B), suggesting that these collagen samples met one of the requirements of the biological evaluation of medical devices, and it can be used in the preparation of cosmetics, drugs, and biomedical materials.

## 3. Materials and Methods

### 3.1. Materials

Air-dried *Megalonibea fusca* swim bladders were obtained from Guangzhou Huangjing Marine Biotechnology Co., Ltd (Guangzhou, China). Reference bovine type I collagen was obtained from the National Institutes for Food and Drug Control. Pepsin derived from porcine stomach mucosa (1:3000 U) was obtained from Macklin (Shanghai, China). Marker standards were received from BBI Life Sciences Corporation (Shanghai, China). All reagents were of analytical grade, unless otherwise indicated.

### 3.2. Chemical Pretreatment 

All collagen extraction was conducted at 4 ℃. Swim bladder pretreatment was conducted based on protocols reported previously [53]. Initially, swim bladders were cut into small pieces and continuously stirred in 0.1 M NaOH at a 1:20 (*w*/*v*) ratio for 24 h to remove pigments and non-collagenous proteins, with the alkaline solution being refreshed every 4 h. Samples were then rinsed repeatedly using dH_2_O until the pH of the water used for washing was neutral. Next, 10% butyl alcohol treatment at a 1:20 (*w*/*v*) ratio was performed for sample defatting, with solvent being replaced every 4 h. After defatting, samples were rinsed 5 times with 10 volumes of cold water, followed by mixing with 85% alcohol at a 1:10 (*w*/*v*) ratio with constant mixing for 4 h to remove sugars. Remaining residues were then washed and cut into small pieces using scissors.

### 3.3. Acid-soluble Collagen Preparation

After pretreatment, swim bladder samples were mixed with 0.5 M acetic acid at a 1:350 (*w*/*v*) ratio for 48 h for ASC extraction, followed by filtering through two layers of gauze and centrifugation for 30 min at 10,000 ×g at 4 ℃. Supernatants were combined with 0.05 M Tris-HCl (pH 7.4), and NaCl was added to a final concentration of 2.6 M for the salt precipitation of proteins. Precipitates were incubated overnight at 4 ℃, followed by centrifugation for 1 h at 15,000 ×g at 4 ℃. Precipitates were then dissolved in a small amount of 0.5 M acetic acid, dialyzed against 10 volumes of 0.5 M acetic acid for 12 h, and against dH_2_O for 48 h, using a membrane with a 10 kDa molecular weight cutoff, with dialysis solution being exchanged every 12 h. The remaining dialysate was then freeze-dried prior to storage at −20 °C [22,53,54].

### 3.4. Pepsin-Soluble Collagen Preparation

The PSC extraction procedures were largely similar to those used for ASC extraction, with some differences. Specifically, swim bladder samples were mixed and extracted with 1% (*w*/*v*) pepsin derived from porcine stomach mucosa in 0.5 M acetic acid at a 1:350 (*w*/*v*) ratio for 48 h, after which the preparation steps were identical to those for ASC [53].

### 3.5. ASC and PSC Yield Analyses

ASC and PSC yields as a fraction of starting dry weight were determined as follows:(1)$$Yield\;\left(\%\right)=\frac{Weight\;of\;freeze-dried\;collagen\;\left(g\right)}{Weight\;of\;dry\;swim\;bladders\;\left(g\right)}\times 100\%$$

### 3.6. SDS-PAGE

SDS-PAGE was performed through a slightly modified version of the Laemmli method [55]. Briefly, ASC, PSC, and reference type I bovine collagen were dissolved at 1 mg/mL using 3% acetic acid, followed by mixing with 2× loading buffer and boiling at 100 ℃ for 1 min. Samples were then loaded onto a prepared gel (4% stacking gel, 7.5% resolving gel). Following electrophoretic separation, proteins were subjected to Coomassie bright blue staining, followed by treatment with a decolorizing solution.

### 3.7. Amino Acid Composition Analyses

Collagen sample amino acid composition was analyzed as per GB 5009.124-2016. Briefly, collagen samples were dissolved in hydrolysis tubes using 6 M HCl, followed by the addition of 6 M HCl. The tubes were then placed in a vacuum, filled with nitrogen, and hydrolyzed for 22 h at 110 ℃. Dry hydrolysate residues were then dissolved using sodium citrate buffer and evaluated with an amino acid analyzer.

### 3.8. UV Absorption Spectrum Analyses

The samples were dissolved at 1 mg/mL using 0.5 M acetic acid and centrifuged for 5 min at 10,000 rpm at 4 ℃, and the supernatants were then analyzed with a UV−Vis spectrometer (Shimadzu UV-2600,Kyoto, Japan) in the 190–400 nm range, with 0.05 M acetic acid as the baseline [41].

### 3.9. FTIR Spectrum Analysis

FTIR analyses of ASC and PSC samples were performed with a Nicolet iS50 FTIR (Thermo Fisher Scientific, Massachusetts, UA) with a horizontal ATR trough plate crystal cell in the 400–4000 cm^−1^ range, collecting automatic signal gains for 32 scans at room temperature at a resolution of 4 cm/s.

### 3.10. Circular Dichroism (CD) 

The conformational characteristics of the extracted collagen samples were assessed by using 0.1 M acetic acid to prepare samples at a final collagen concentration of 0.2 mg/mL and then transferring these samples into a CD spectrometer (Applied Photophysics Ltd Chirascan plus, surrey, UK) sample pool with a 1 mm optical path to conduct a scan from 190–260 nm. Scans were taken at a rate of 100 nm/min, with a 1 nm interval at 25 ℃.

### 3.11. X-ray Diffraction (XRD)

An X-ray diffractometer (Rigaku D/Max-2200, Tokyo, Japan) was used to assess the freeze dried ASC and PSC diffraction patterns at 40 kV, with a current of 40 mA and CuKα radiation (λ = 1.5406 Å). The XRD pattern for each sample was then assessed with the Bragg equation:(2)$$d=\frac{\lambda }{2sin\theta }$$
where *d* is the distance between adjacent crystal planes, *θ* is the Bragg angle, and *λ* is the X-ray wavelength [56].

### 3.12. Microstructural Analysis 

Collagen and swim bladder samples were transferred to the loading platform and subjected to ion spattering to increase conductivity. SEM (Hitachi S-2380N, Tokyo, Japan) was then used to perform microstructural analyses at an accelerating voltage of 5.0 kV.

For TEM and AFM analyses, collagen fibrils were prepared using a slightly modified version of a protocol reported previously by Yousefi et al. [47]. Briefly, 0.02 mM HCl was used to dissolve the collagen samples, followed by dilution with phosphate buffer at a 9:1 ratio. Collagen fibrils were formed by warming these samples for 3 h at 30 ℃, after which a single drop of the solution was loaded onto copper mesh (200-mesh), followed by 2% (*w*/*v*) phosphotungstic acid treatment for TEM sample preparation. For AFM analyses, samples were poured onto mica sheets, allowed to air dry, rinsed two times with water, and allowed to air dry again. The collagen samples were then analyzed via TEM (JEOL JEM 2100F, Tokyo, Japan) or AFM (Bruker Bioscope Catalyst/Muitimode, Billerica, MA, USA).

### 3.13. Thermal Analysis

Approximately 3–5 mg of individual samples were weighed and transferred into an aluminum DSC crucible that was then sealed. DSC was then used to quantify the thermal transition (T_max_) using an appropriate instrument (NETZSCH DSC204F1, Selb, Germany), with an empty crucible as a reference control. The samples were heated from 0–75 ℃ at 1 ℃/min.

### 3.14. Zeta Potential Measurement

Freeze-dried collagen samples were dissolved with 0.5 M acetic acid to a final concentration of 2 mg/mL, after which HCl or NaOH (1 M) was used to adjust the pH to final values ranging from 2–8, with zeta (ζ) potential and pI then being quantified with a zeta potential analyzer (Malvern Zetasizer Nano ZS, Malvern, UK).

### 3.15. Analysis of Collagen Solubility at Different pH Levels and NaCl Concentrations 

ASC and PSC solubility was assessed as reported previously by Yu et al. [57]. Briefly, samples were dissolved at 3 mg/mL in 0.5 M acetic acid, and 7 mL of the resultant solution was added to a 10 mL beaker after which the final pH was adjusted using HCl (6 M) or NaOH (6 M) for final values of 2, 4, 6, 8, or 10, with pH-matched dH_2_O subsequently added to a final volume of 10 mL. The samples were then mixed slowly for 10 min, centrifuged for 20 min at 10,000× *g* at 4 ℃, and the relative solubility was measured as the ratio of collagen solubility to the maximum collagen solubility at each pH level.

Alternatively, after dissolution in 0.5 M acetic acid at a final concentration of 6 mg/mL, these samples were adjusted to a final NaCl concentration of 0, 10, 20, 30, 40, 50, or 60 mg/mL, respectively. Samples were then mixed slowly for 10 min, centrifuged for 20 min at 10,000× *g* at 4 ℃, and the relative solubility was measured, with a solution containing 0 mg/mL NaCl serving as the reference control.

### 3.16. Cytotoxicity Analysis

Collagen cytotoxicity was assessed via a CCK-8 assay using BALB/C-3T3 fibroblasts and human keratinocytes (HaCaT cells), as per GB/T 16886.5–2017. Briefly, samples of collagen (<0.5 mm thick, 3 cm^2^/mL) were extracted for 72 h in cell culture media at 37 ℃. The cells were thenplated in 96-well plates (1.5 × 10^4^/well) and incubated for 24 h, after which they were treated with a range of concentrations of the prepared collagen extracts (0, 6.25, 12.5, 25, 50, or 100%), followed by incubation for 24 h in a humidified 5% CO_2_ tissue culture incubator. CCK-8 solution was then added into each well, followed by an additional 2.5 h incubation. The absorbance in each well was then assessed at 450 nm.

## 4. Conclusions

In summary, ASC and PSC samples were successfully extracted from *Megalonibea fusca* swim bladders, with PSC yields being higher than those of ASC. While pepsin was able to enhance collagen extraction efficiency, it had no adverse impact on the triple-helical structure of the isolated collagen. Both the ASC and PSC samples were found to consist of typical type I collagen, with the expected triple helix structure. While the physicochemical characteristics of these two collagen extracts differed slightly, neither showed cytotoxicity. *Megalonibea fusca* swim bladders thus represent a promising potential alternative source of type I collagen for commercial use.

## Figures and Tables

**Figure 1 marinedrugs-21-00159-f001:**
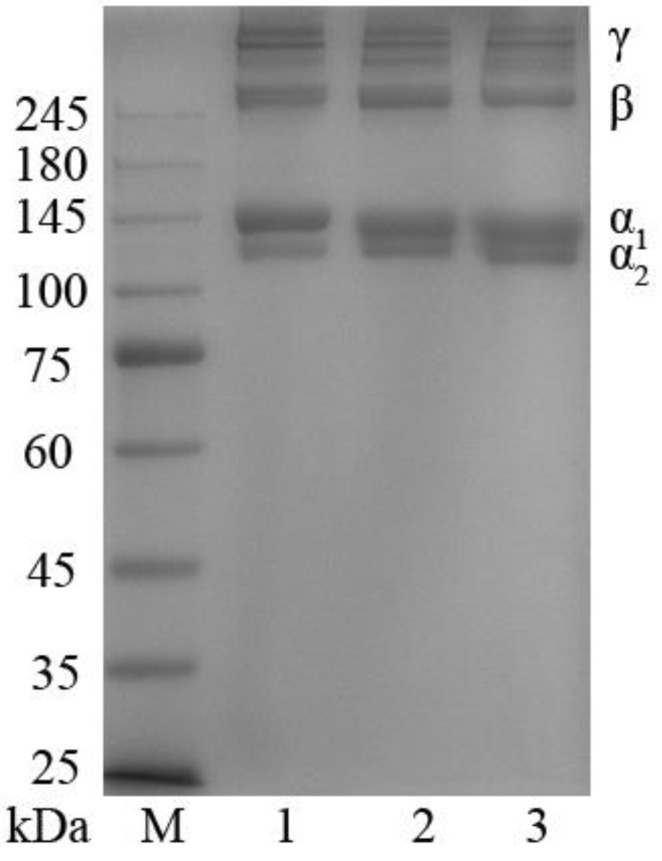
SDS-PAGE separation patterns for ASC and PSC samples extracted from the swim bladders of *Megalonibea fusca*. Lane 1: Bovine Type I reference material; Lane 2: ASC; Lane 3: PSC.

**Figure 2 marinedrugs-21-00159-f002:**
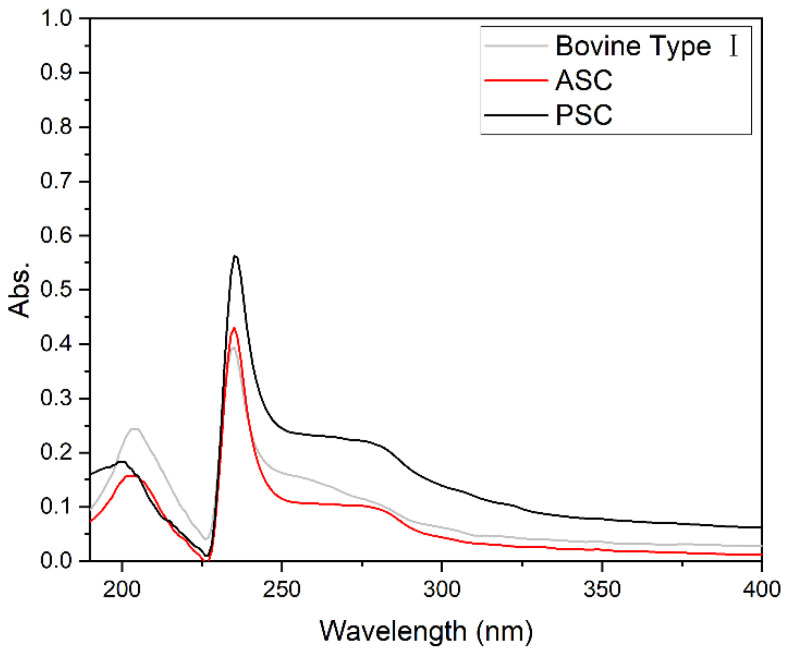
UV absorption spectra for ASC and PSC samples extracted from the swim bladders of *Megalonibea fusca* and reference bovine type I collagen.

**Figure 3 marinedrugs-21-00159-f003:**
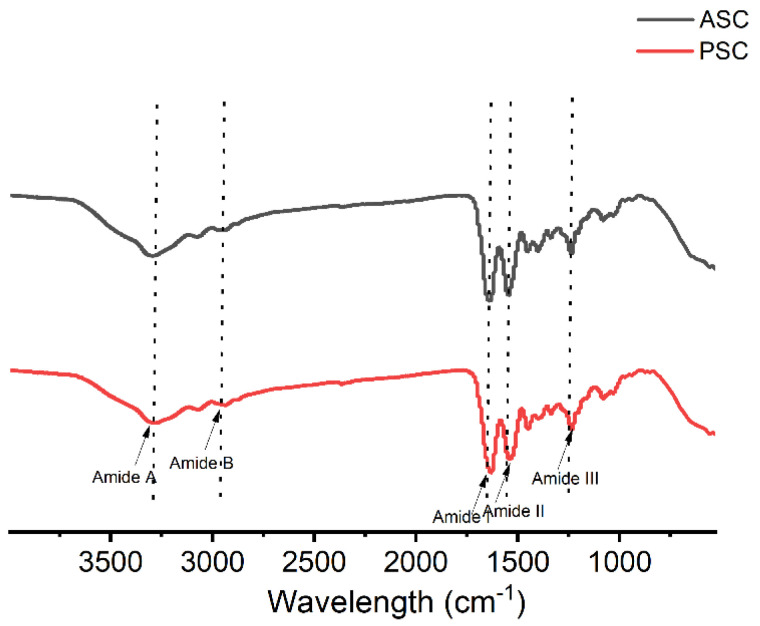
The FTIR spectra of ASC and PSC.

**Figure 4 marinedrugs-21-00159-f004:**
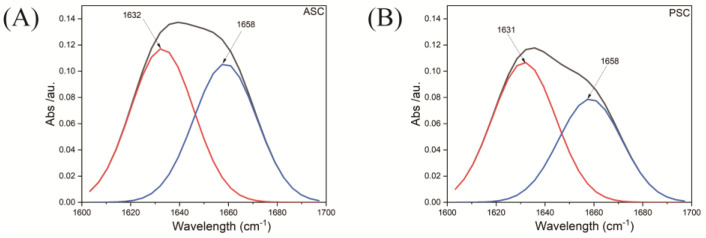
Amido-I peak deconvolution curves for ASC and PSC.

**Figure 5 marinedrugs-21-00159-f005:**
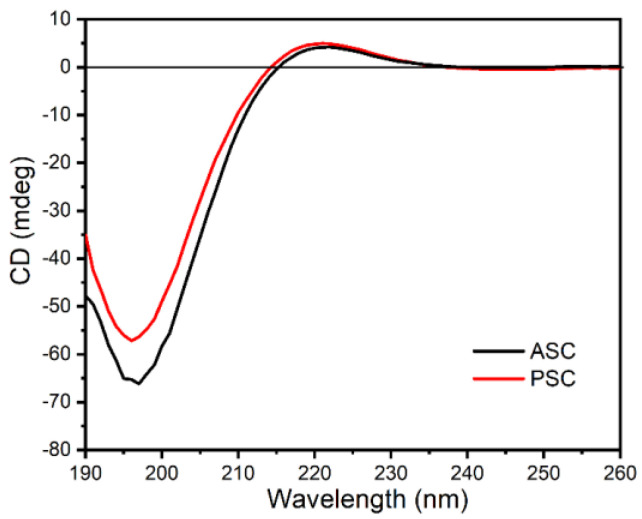
The CD spectra of ASC and PSC.

**Figure 6 marinedrugs-21-00159-f006:**
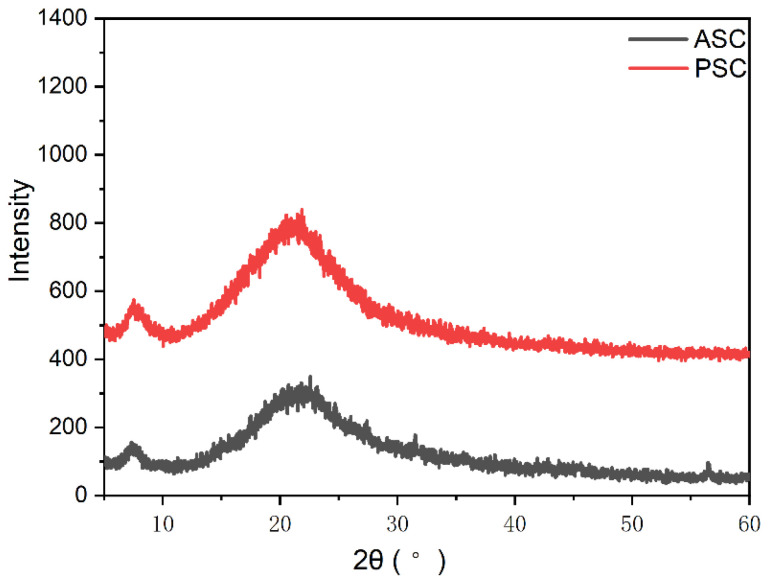
X-ray diffraction results for ASC and PSC samples.

**Figure 7 marinedrugs-21-00159-f007:**
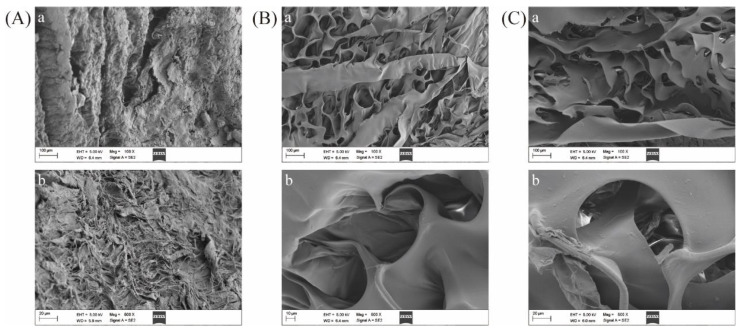
Scanning electron microscopy images of *Megalonibea fusca* swim bladders (**A**), ASC (**B**), and PSC (**C**). SEM images are shown at different levels of magnification.

**Figure 8 marinedrugs-21-00159-f008:**
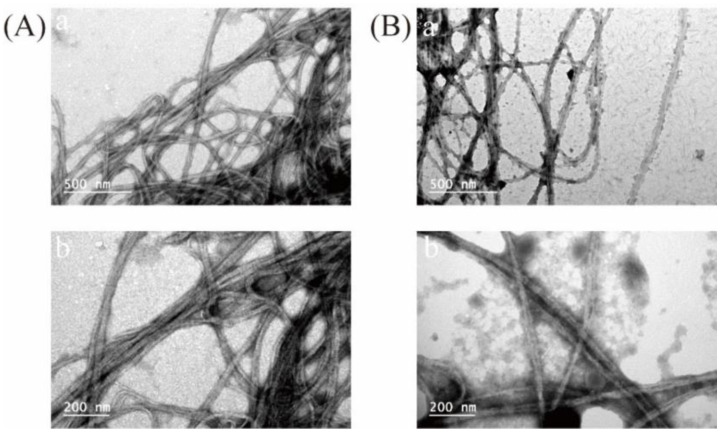
Transmission electron microscopy images of ASC (**A**) and PSC (**B**). TEM images are shown at different levels of magnification.

**Figure 9 marinedrugs-21-00159-f009:**
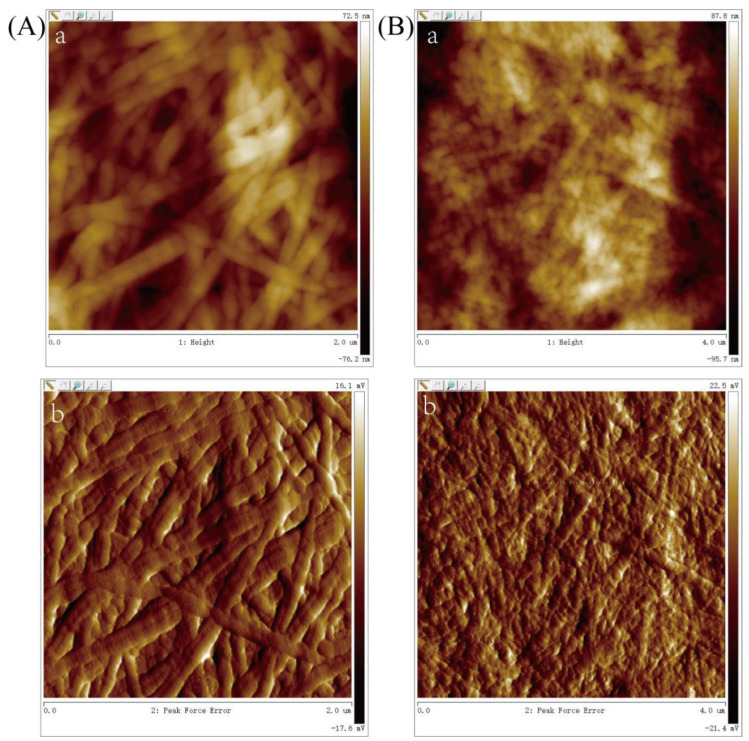
Atomic force microscopy images of ASC (**A**) and PSC (**B**), with corresponding phase (**a**) and height (**b**) AFM images.

**Figure 10 marinedrugs-21-00159-f010:**
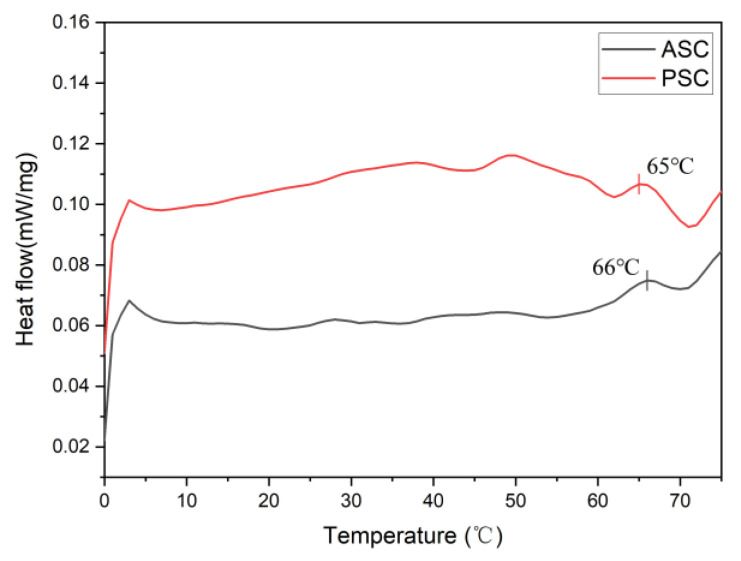
Differential scanning calorimetry (DSC) curves for ASC and PSC.

**Figure 11 marinedrugs-21-00159-f011:**
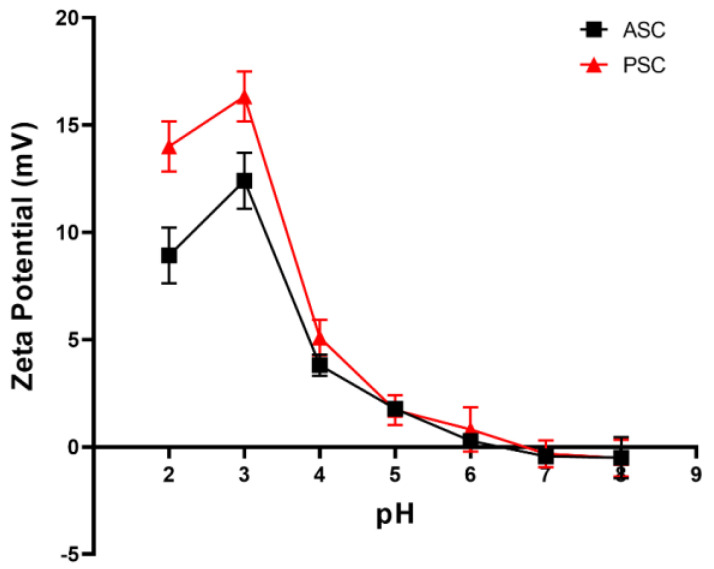
ASC and PSC sample zeta potential values.

**Figure 12 marinedrugs-21-00159-f012:**
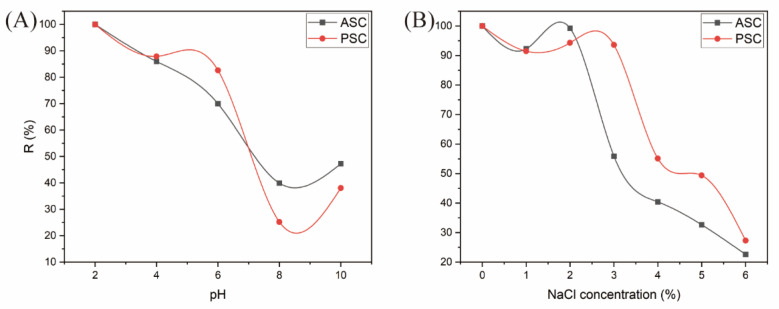
ASC and PSC sample relative solubility (%) at varying pH levels (**A**) and NaCl concentrations (**B**).

**Figure 13 marinedrugs-21-00159-f013:**
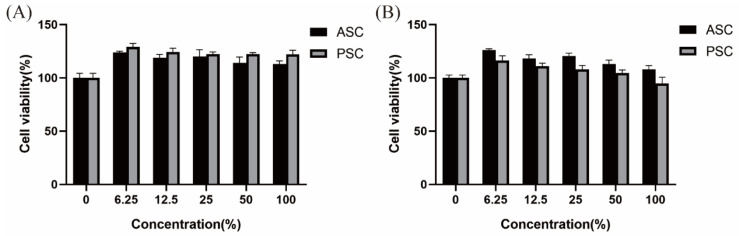
CCK-8 assay results corresponding to the cytotoxic effects of ASC and PSC when used to treat BALB/C-3T3 (**A**) and HaCaT cells (**B**).

**Table 1 marinedrugs-21-00159-t001:** ASC or PSC yields in extracts from the swim bladders of different species of fish.

Species	Yields (%)	
	ASC	PSC
Catla [19]	22.20	61.30
Bester sturgeon [20]	N/A	37.70
Rohu [21]	N/A	46.52
Bighead carp [23]	N/A	59.00
Miiuy croaker [15]	1.33	8.37
yellowfin tuna [22]	1.07	12.10
*Megalonibea fusca*	33.38	84.79

**Table 2 marinedrugs-21-00159-t002:** The amino acid content of ASC and PSC samples extracted from the swim bladders of *Megalonibea fusca*. The results are expressed as residues/1000 total amino acid residues.

Amino Acids	Residues/1000 Total Amino Acid Residues	
	ASC	PSC
Asp	44	40
Thr	28	28
Ser	33	34
Glu	78	77
Gly	316	320
Ala	151	153
Val	22	21
Met	12	12
Ile	8	8
Leu	21	20
Tyr	4	3
Phe	11	10
Lys	25	24
His	3	3
Arg	49	48
Pro	104	106
Hyp	91	93
Imino acid	195	199

**Table 3 marinedrugs-21-00159-t003:** The deconvolution of the amido I band and the relative amount of the triple helix of collagen from *Megalonibea fusca* swim bladders.

Collagen	1631 cm^−1^	1658 cm^−1^	Triple Helix (%)
ASC	3.7817932	3.4108148	0.525788865
PSC	3.4625435	2.5575261	0.575166689

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
