# Peer review of "Extraction and Characterization of Pepsin- and Acid-Soluble Collagen from the Swim Bladders of Megalonibea fusca"

_marinedrugs, 2023, doi:10.3390/md21030159_

Round 1

Reviewer 1 Report

This work has presented the extraction of collagen from swim bladders of Megalonibea fusca then full characterisation of the extracted collagen using several techniques. This work seems novel and original however authors must address the below comments.

1.    L74: Can authors add the reference for each row in table 1 so will be easy for the reader to follow.

2.     Can authors explain the small decrease in TYR from 4 in ASC to 3 in PSC although the pepsin has cleaved the telopeptide and seen in the SDS figure 1. TYR is found in telopeptide and 3 TYR residues in PSC should suggest pepsin has not removed all telopeptides which does not correlate to the decrease in the MWt seen in figure 1.

3.    Can authors explain the PHE detected in collagen as collagen does NOT contain PHE? this suggest collagen purity is low and sample might contain non-collagen fractions which again does not match with the clean SDS in figure 1. It is important to have accurate amino acid profile. Also can authors explain the high MET amount and can they explain the absent of CYS?

4.    L174: Can authors explain the D-periodicity in PSC although pepsin has removed the telopeptide which is essential for the formation of D-periodicity. Can authors also explain how the D-periodicity is similar in ASC and PSC although telopeptide removed from PSC.  Can authors provide details on how the D-periodicity was measured and provide the number of samples used for that. Can authors include the RSD or CVs for each measure?

5.    L195: The difference in the HYP does not explain the large difference in Td. Can authors provide more replicates for the thermal analysis with RSD%. This will help to use the average and rule out outliers.

6.    L295: Can authors provide more details such as scan rate and temperature.

Author Response

Response to Reviewer 1 Comments

Dear Reviewer,

Thanks very much for taking your time to review this mamuscript marinedrugs-2209268. I really appreciate all your comments and suggestions! Please find my itemized responses in below and my correction in the re-submitted files.

Thanks again!

Point 1: L74: Can authors add the reference for each row in table 1 so will be easy for the reader to follow.

Response 1: Thanks for your careful checks. Baesd on your comments, we have made the correction.

Point 2: Can authors explain the small decrease in TYR from 4 in ASC to 3 in PSC although the pepsin has cleaved the telopeptide and seen in the SDS figure 1. TYR is found in telopeptide and 3 TYR residues in PSC should suggest pepsin has not removed all telopeptides which does not correlate to the decrease in the MWt seen in figure 1.

Response 2: We think that  the telopeptide does not  just contain TRY. The removal of telopeptide by pepsin causes MWt to decrease and it does not need to be completely removed.

Point 3: Can authors explain the PHE detected in collagen as collagen does NOT contain PHE? this suggest collagen purity is low and sample might contain non-collagen fractions which again does not match with the clean SDS in figure 1. It is important to have accurate amino acid profile. Also can authors explain the high MET amount and can they explain the absent of CYS?

Response 3: We think PHE was detected because the sample may contain non-collagen components, but the non-collagen component was so low that the SDS picture was clean. The high MET amount might also be related to sample purity. And the absence of CYS is that we have no detection conditions.

Point 4: L174: Can authors explain the D-periodicity in PSC although pepsin has removed the telopeptide which is essential for the formation of D-periodicity. Can authors also explain how the D-periodicity is similar in ASC and PSC although telopeptide removed from PSC.  Can authors provide details on how the D-periodicity was measured and provide the number of samples used for that. Can authors include the RSD or CVs for each measure?

Response 4: The D-periodicity of ASC and PSC was obtained by analyzing the transmission electron microscope image by imageJ, and the RSDs about the D-periodicity of ASC and PSC were 4.7% and 6.2%.

Point 5: L195: The difference in the HYP does not explain the large difference in Td. Can authors provide more replicates for the thermal analysis with RSD%. This will help to use the average and rule out outliers.

Response 5: We repeated the DSC experiment and corrected the experimental results.

Point 6: L295: Can authors provide more details such as scan rate and temperature.

Response 6: Thanks for your careful checks. Baesd on your comments, we have made the correction.

Reviewer 2 Report

This investigation proposes a source of collagen from fish bladder from Megalonibea fusca as an alternate to mammalian collagen.

Overall the investigation has carried out biochemical and biophysical analysis to characterise the extracted collagens.  Many analytical methods were undertaken with little explanation of why they were performed, instead referred to other work (references)

The manuscript needs to address major flaws in interpretation of results and presentation of figures and figure legends.

The general method descriptors and explanations need revising to reflect correct known theory of collagens. More clarity on how materials were prepared for analysis is also identified.

The English language needs to be corrected throughout the manuscript.  All sections have sentences and grammar errors rendering the text difficult to read.   Whole document needs to be read and English language revised

The following are points to address; 

Abstract

Q1 Line 25 – biological evaluation of medical devices encompasses many different tests, one of which is cytotoxicity.  Therefore it would be correct to say ‘ …which met one of the requirements of biological evaluations of medical devices’

Introduction

Q2 English language needs correction throughout the document and in this section for improved readability.

Some examples, 

Line 41…..’the processing byproducts’  to the processing of byproducts’

Line 44   ‘And the special factors, such as religion…’  to  ‘In addition,  such as religion  

Q3 Line 53/54, needs rephrasing

‘However, the research on Megalonibea fusca is only artificial breeding and other studies are mostly blank.’

Results and Discussion

2.1 Yield

Q4 Table 1 , please include results of the Megalonibea fusca yield as well in the table

2.2 SDS PAGE

Q5 Line 78   What is meant by ‘fish maw’?

Q6 The molecular weight markers are likely from standard globular proteins commercially sourced but have not been described in the methods.

One cannot derive collagen molecular weights in comparison to globular proteins as collagen chains migrate on SDS PAGE very differently to globular proteins as they are more linear with higher aspect ratios.

Authors are referred to the following

The anomalous mobility of collagen on SDS-PAGE has been known for decades.  See, for example,  Svojtkova E, Deyl Z, Adam M. Anomalies in the electrophoretic molecular-weight determination of collagen denaturation products. Journal of Chromatography A. 1973 Sep 26;84(1):147-53,  and subsequent papers by others.

2.6 Circular Dichroism

Q7 Line 140 Revise this sentence to reflect the reason for the analysis.  CD is generally used for collagens as an indication of intact tertiary structure.

Q8 What temperature was the CD carried out at? Please state this in the methods as fish and mammalian collagen have different melting temperatures in solution which may have accounted for lower triple helix signal.

Q9  Fish samples should be measured with CD alongside Bovine collagen for comparison at the same temperature.

Q10 Table 3 incorrect title

Q11 Line 146  What is meant by ‘naturally active collagen’ do the authors mean ‘ native collagen’ ?  

2.7 X ray diffraction

Q12 preparation of samples is missing from method, please describe

Q13 not clear why the researchers carried out this part of the work

Q14 Line 152 ‘crystal structure’, not accurate to describe this as it is a fibre-based material which doesn't show the complex, sharp lattice reflections seen from crystals of globular proteins.  It is a fibre diffraction pattern.

2.8. Microstructural Analysis

Q15  Incorrect reference to Figure, should be Figure 7 in text “ Figure 6 shows the microstructures of swim bladder (Figure 7A), ASC (Figure 7B)’

 Q16 Figure 8 Should state ‘Transmission electron microscopy’ not ‘Scanning’

Q17 Line 173/174, the diameter is ‘wider’ not ‘longer’,

Q18 Figure 9 scale bars are missing

Q19 Figure 9 what is meant by the drawings of Atomic force microscopy, is this missing in the figure?

2.9. Thermal analysis

Q20 Method is not clear if it was performed on freeze dried material or a solution.  This needs to be included in a method.

Q21 The explanation of why PSC is higher in its Tm is not acceptable as it is highly unlikely that a difference in 4 Hydroxyproline residues would impact on 18 C difference.   How many replicates of each sample type were analysed and what was the error %.   It is concerning to see these results and a repeat experiment should be performed to ensure there are no differences in sample treatments and preparation.

2.12. Cytotoxicity

Q22 Figure 12 referred in text, should be Figure 13

23# Refer to comment in abstract around biological evaluation

3. Materials and Methods

3.1. Materials

Q24 Line 240 when is collagenase used in the methods?

Author Response

Response to Reviewer 2 Comments

Dear Reviewer,

Thanks very much for taking your time to review this mamuscript marinedrugs-2209268. I really appreciate all your comments and suggestions! As for English language and style, I have revised the content of the entire manuscript. Please find my itemized responses in below and my correction in the re-submitted files.

Thanks again!

Abstract

Point 1: Line 25 – biological evaluation of medical devices encompasses many different tests, one of which is cytotoxicity.  Therefore it would be correct to say ‘ …which met one of the requirements of biological evaluations of medical devices’

Response 1: Thanks for your careful checks. Baesd on your comments, we have made the correction.

Introduction

Point 2: English language needs correction throughout the document and in this section for improved readability.

Some examples,

 Line 41….. ’the processing byproducts’  to the processing of byproducts’

 Line 44   ‘And the special factors, such as religion…’  to  ‘In addition,  such as religion   “

Response 2: Thanks for your careful checks. Baesd on your comments, we have made the correction.

Point 3: Line 53/54, needs rephrasing

‘However, the research on Megalonibea fusca is only artificial breeding and other studies are mostly blank.’

Response 3: Thanks for your careful checks. Baesd on your comments, we have made the correction.

Results and Discussion

2.1 Yield

Point 4:Table 1 , please include results of the Megalonibea fusca yield as well in the table

Response 4:Thanks for your careful checks. Baesd on your comments, we have made the correction.

2.2 SDS PAGE

Point 5:Line 78   What is meant by ‘fish maw’?

Response 5: ‘fish maw’ is just another way of saying ‘swim bladder’.We have made the correction to make the word harmonized within the whole manuscript.

Point 6:The molecular weight markers are likely from standard globular proteins commercially sourced but have not been described in the methods.

One cannot derive collagen molecular weights in comparison to globular proteins as collagen chains migrate on SDS PAGE very differently to globular proteins as they are more linear with higher aspect ratios.

Authors are referred to the following

The anomalous mobility of collagen on SDS-PAGE has been known for decades.  See, for example,  Svojtkova E, Deyl Z, Adam M. Anomalies in the electrophoretic molecular-weight determination of collagen denaturation products. Journal of Chromatography A. 1973 Sep 26;84(1):147-53,  and subsequent papers by others.

Response 6: All we need is to do a qualitative analysis of the extracted collagen and compare the molecular weight difference between ASC and PSC. We also added the Bovine Type I reference material for comparison.

2.6 Circular Dichroism

Point 7:Line 140 Revise this sentence to reflect the reason for the analysis.  CD is generally used for collagens as an indication of intact tertiary structure.

Response 7: Thanks for your careful checks. Baesd on your comments, we have made the correction.

Point 8:What temperature was the CD carried out at? Please state this in the methods as fish and mammalian collagen have different melting temperatures in solution which may have accounted for lower triple helix signal.

Response 8: Thanks for your careful checks. The CD carried out at 25℃.

Point 9:Fish samples should be measured with CD alongside Bovine collagen for comparison at the same temperature.

Response 9: We just wanted to use CD to characterize the triple helix structure of collagen, so we did not use Bovine collagen for comparison

Point 10:Table 3 incorrect title

Response 10: Thanks for your careful checks. Baesd on your comments, we have made the correction.

Point 11:Line 146  What is meant by ‘naturally active collagen’ do the authors mean ‘ native collagen’ ?  

Response 11: I made a mistake in my choice of words. We have made the correction.

2.7 X ray diffraction

Point 12: preparation of samples is missing from method, please describe

Response 12: The samples are freeze-dried ASC and PSC. We have added to the method

Point 13:not clear why the researchers carried out this part of the work

Response 13: We learned that X-ray diffraction is considered to be one of the main methods to detect the spatial structure and triple helix conformation of collagen

Point 14:Line 152 ‘crystal structure’, not accurate to describe this as it is a fibre-based material which doesn't show the complex, sharp lattice reflections seen from crystals of globular proteins.  It is a fibre diffraction pattern.

Response 14: Thanks for your careful checks. Baesd on your comments, we have made the correction.

2.8. Microstructural Analysis

Point 15:Incorrect reference to Figure, should be Figure 7 in text “ Figure 6 shows the microstructures of swim bladder (Figure 7A), ASC (Figure 7B)’

Response 15: Thanks for your careful checks. Baesd on your comments, we have made the correction.

Point 16:Figure 8 Should state ‘Transmission electron microscopy’ not ‘Scanning’

Response 16: Thanks for your careful checks. Baesd on your comments, we have made the correction.

Point 17:Line 173/174, the diameter is ‘wider’ not ‘longer’,

Response 17: Thanks for your careful checks. Baesd on your comments, we have made the correction.

Point 18:Figure 9 scale bars are missing

Response 18: The scale of Figure 9 is indicated on the figure. Probably because the picture is too small to show clearly

Point 19:Figure 9 what is meant by the drawings of Atomic force microscopy, is this missing in the figure?

Response 19: I made a mistake in my choice of words. We have made the correction.

2.9. Thermal analysis

Point 20:Method is not clear if it was performed on freeze dried material or a solution.  This needs to be included in a method.

Response 20: Thanks for your careful checks. Baesd on your comments, we have made the correction.

Point 21:The explanation of why PSC is higher in its Tm is not acceptable as it is highly unlikely that a difference in 4 Hydroxyproline residues would impact on 18 C difference.   How many replicates of each sample type were analysed and what was the error %.   It is concerning to see these results and a repeat experiment should be performed to ensure there are no differences in sample treatments and preparation.

Response 21: We repeated the DSC experiment and corrected the experimental results.

2.12. Cytotoxicity

Point 22:Figure 12 referred in text, should be Figure 13

Response 22:  I made a mistake in my choice of words. We have made the correction.

Point 23:Refer to comment in abstract around biological evaluation

Response 23: I made a mistake in my choice of words. We have made the correction.

  1. Materials and Methods

Point 24:Line 240 when is collagenase used in the methods?

Response 24: The collagenase is not used in the methods. We have made the correction.

Reviewer 3 Report

I have only one comment regarding to the procedure of preparation of PSC. Is the ratio during PSC extraction correct? 1:350 (w/v) seems to be too high.

Author Response

Response to Reviewer 3 Comments

Dear Reviewer,

Thanks very much for taking your time to review this mamuscript marinedrugs-2209268. I really appreciate all your comments and suggestions! Please find my itemized responses in below and my correction in the re-submitted files.

Thanks again!

Point 1:  Is the ratio during PSC extraction correct? 1:350 (w/v) seems to be too high.

Response 1: Thanks for your careful checks. The ratio during PSC extraction is correct. Because the swim bladder I used was air-dried, it lost 70 percent of its moisture compared to fresh fish.
